# High-Efficiency Reconfigurable CMOS RF-to-DC Converter System for Ultra-Low-Power Wireless Sensor Nodes with Efficient MPPT Circuitry

Roberto La Rosa [1,2,*], Danilo Demarchi [3], Sandro Carrara [4] and Catherine Dehollain [2]

1 STMicroelectronics, Stradale Primosole 50, 95121 Catania, Italy
2 Ecole Polytechnique Federale de Lausanne, CH-1015 Losanne, Switzerland; catherine.dehollain@epfl.ch
3 Department of Electronics and Telecommunications, Politecnico di Torino, C.so Duca degli Abruzzi 24, 10129 Torino, Italy; danilo.demarchi@polito.it
4 Laboratory of Integrated Circuits, École Polytechnique Fédérale de Lausanne, CH-2002 Neuchâtel, Switzerland; sandro.carrara@epfl.ch
* Correspondence: roberto.larosa@st.com; Tel.: +11-39-347-293-6286

**Abstract:** This paper presents a novel CMOS RF-to-DC converter for ultra-low-power wireless sensor nodes powered by RF wireless power transfer. The proposed converter achieves 10% higher power conversion efficiency than a conventional rectifier, with only a 1% increase in power consumption. The system employs a reconfigurable Dickson topology, operates on the unlicensed 868 MHz ISM band, and includes a built-in power-efficient MPPT system architecture. Experimental measurements show a maximum power conversion efficiency of 55% in the power range from $-22\,\text{dBm}$ to $0\,\text{dBm}$, with a power sensitivity of $-22\,\text{dBm}$ for a DC output voltage of 2.4 V. The proposed converter offers a promising solution for efficient wireless power transfer and energy harvesting in ultra-low-power wireless sensor nodes.

**Keywords:** CMOS technology; maximum power point tracking (MPPT); RF-to-DC converter; RF energy harvesting; wireless power transfer; ultra-low-power; wireless sensor nodes; power conversion efficiency





## 1. Introduction

Wireless Sensor Nodes (WSNs) are becoming more widespread due to their increasing complexity and ability to perform sensing, data analysis, and communication [1]. However, powering these nodes efficiently and effectively presents a challenge. Conventional nodes typically rely on batteries, which can lead to increased costs, maintenance requirements, and difficulties in miniaturization [2]. Battery-free devices are becoming increasingly popular among engineers as a convenient option, especially in applications such as high-temperature or hazardous locations, wearable devices, or biomedical applications [3]. For these reasons, designers are showing more interest in RF Wireless Power Transfer (WPT) and energy harvesting (EH) technologies.

RF WPT is gaining attention due to its ability to reach non-visible locations and its widespread availability. Unlike other energy sources, such as photovoltaic and kinetic, which are highly dependent on weather conditions, RF WPT takes advantage of the high prevalence of RF signals and their lower time dependence [4]. However, implementing efficient RF WPT can be complex, and designing a low-input-power RF-to-DC converter is always challenging. Despite the prevalence of RF signals, the available RF power is typically low, making it only possible to power electronic devices with ultra-low power consumption (several microwatts). This limited RF power is due to space path loss and regulations that limit RF power emissions to protect human health [5]. Despite restricted transmission power and space path loss, optimizing the design of RF-to-DC converters

for power conversion efficiency (PCE) and power sensitivity can convert received RF power into sufficient electrical power to supply electronic devices [6]. Power sensitivity refers to the minimum RF power required for the converter to start harvesting energy into a storage device. Designers aim to minimize power sensitivity to harvest energy from distant and low-power sources. Various design methods, techniques, and circuit topologies for RF-to-DC converters have been proposed in the literature to achieve optimal power sensitivity performance [7–11].

Many of these papers propose techniques for efficiently biasing the transistors that rectify the low-voltage input signal, with a focus on compensating for the threshold voltage $V_T$ of the gate-source voltage of the transistor, as this directly affects the rectifier's efficiency. For example, the authors of [12] proposed a self-compensation method that extends the length of the compensating bridges to increase the gate bias offset, while the authors of [13] used a static and dynamic self-compensation technique to decrease the threshold voltage of the rectifying transistors. The authors of [14] report a dual-topology CMOS rectifier with a peak PCE of 78.4% operating at 900 MHz. While these publications have proposed improvements for power sensitivity performance regardless of the RF-to-DC converter topology, they do not consistently contribute to the further optimization of PCE performance when the input-received RF power is increased beyond the power sensitivity. As a result, conventional RF-to-DC converters typically exhibit decreasing PCE performance as input power increases, which can be inefficient in WPT applications that deal with variable power conditions, such as asset tracking tags and over-the-distance wireless battery chargers, where the relative distance between the power transmitter and receiver is not constant [6,15]. To address this problem, several authors have proposed reconfigurable RF-to-DC converter architectures to increase output power and improve the PCE over a wide range of input power levels. For example, the authors of [16] presented a reconfigurable RF-to-DC converter, while the authors of [17] proposed a maximum power point tracking (MPPT) technique to optimize the number of rectifiers in a reconfigurable RF-to-DC converter for high PCE. Similarly, the authors of [18] presented a reconfigurable RF-to-DC converter to improve PCE performance. In [19], the authors describe a RF energy-harvesting system based on a reconfigurable rectifier designed to maintain high PCE over a wide range of input power levels, achieving a maximum PCE of 39% from $-22\,\text{dBm}$ to $-2\,\text{dBm}$. The authors of [20] showcased a solution for MPPT, where they utilized an auxiliary RF rectenna to provide a RF power-dependent reference voltage (Vref). This Vref was then used to harvest the maximum available power from the main rectenna for a range of received RF power from $-11\,\text{dBm}$ to $3\,\text{dBm}$. The authors of [21] present a novel circuit architecture for a Dickson-based reconfigurable rectifier with a wide power dynamic range, achieving a PDR of 14 dB and a peak PCE of 34.93% operating at 900 MHz.

This work proposes an RF-to-DC converter architecture operating in the 868 MHz ISM band that implements an MPPT system. The MPPT system allows for maximizing the power conversion efficiency of a RF WPT system over a wide range of input power levels that is achieved through a converter with a configurable number of rectifier stages and an innovative open-circuit voltage measurement technique. The result is a cost-effective and silicon area-effective MPPT system that increases the PCE of the system, regardless of the voltage rectifier topology. The system combines reconfigurable voltage rectifiers with indirect monitoring of input-received power using the open-circuit voltage of an unloaded voltage rectifier. This study presents the underlying theory, layout implementation, and experimental results for the CMOS implementation.

This work introduces several innovative aspects, including the following:

- Development of a system-level strategy for correctly biasing the operating point of the RF-to-DC converter, which is a novel concept not commonly found in existing architectures. This strategy focuses on optimizing the output voltage of the rectifier by properly biasing the system, rather than just the rectifier itself. This approach provides a more effective way to maximize power conversion efficiency with no impact on the power sensitivity performance, with negligible impact on circuit complexity and

effective die area. The proposed method and guidelines provide a practical solution for achieving optimal performance in RF-to-DC converter architectures.

- Implementation of an innovative ultra-low power voltage measurement technique that uses a simple rectifier to indirectly monitor input-received power, which is a novel concept not commonly found in existing RF-to-DC converter architectures and provides a more accurate and efficient way to optimize power conversion efficiency. This technique uses a very small silicon area, making it a cost-effective solution for practical applications.
- Combination of reconfigurable voltage rectifiers with indirect monitoring of input-received power to create a cost-effective and silicon area-effective MPPT system that increases the PCE of the system, regardless of the voltage rectifier topology. The proposed MPPT system uses an ultra-low power circuit with minimal impact in terms of extra power consumption and effective die area increase, making it a practical solution for energy harvesting applications.

Section 2 describes the architecture of a conventional WPT system. Section 3 explains the proposed MPPT technique and shows experimental results demonstrating the achieved PCE improvement compared to a conventional RF-to-DC converter architecture along with the measurement setup and experimental results. Section 4 provides a summary and discussion of the achieved results.

## 2. Materials and Methods

Figure 1 shows a simplified and generic block diagram of a WPT RF power conversion system that, in its minimal architecture, comprises the following building blocks:

- The RF power transmitter is responsible for transmitting a maximum RF power in compliance with regulations that ensure human health is not compromised.
- The Matching Network (MN) matches the impedance of the antenna with that of the RF-to-DC converter to optimize the power transfer.
- The RF-to-DC converter rectifies and amplifies the input voltage.
- The storage capacitor $C_{storage}$ stores the harvested energy.
- The ultra-low-power management system regulates and controls the voltage supplied to the load, which is typically a wireless sensor node.

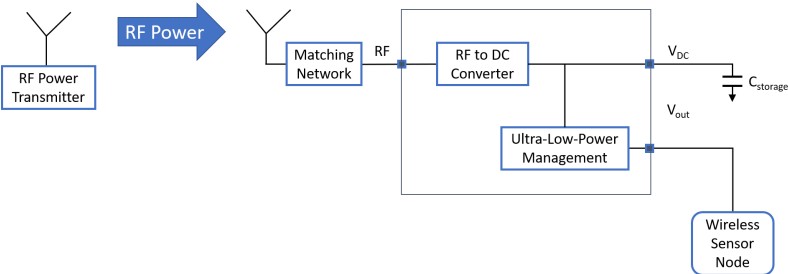

**Figure 1.** WPT system block diagram.

The reference performance metrics for designing the RF-to-DC converter include the minimum distance $d_{min}$ to cover between the power transmitter and receiver, the sensitivity power level $P_{RF\_min}$, the DC output voltage $V_{DC}$, the DC output power $P_{DC\_min}$, and the operating frequency.

Regarding the minimum distance $d_{min}$, the System on Chip (SoC) is defined for an application to cover a distance of more than 7 m between the power transmitter and receiver under the following conditions:

- Free-space WPT.
- Unity gain for both transmitter and receiver antennas.
- Transmitted power = 27 dBm.
- Frequency = 868 MHz.

Concerning the sensitivity power level, i.e., the minimum output power RF_min to achieve the minimum distance $d_{min}$ in the conditions expressed above, the following specifications are set for the RF-to-DC converter:

- $P_{RF\_min}$ = −22 dBm = 6.3 µW
- $PCE_{min}$ = 40%

Regarding the DC output voltage $V_{DC}$ and the minimum DC output power $P_{DC\_min}$ to meet the specification of powering other standard components with a typical supply voltage of 1.8 V, the following specifications are set for the RF-to-DC converter:

- Output voltage $V_{DC\_oc}$ = 4.8 V (Open Circuit condition).
- Output Voltage $V_{DC}$ = 2.4 V (at $I_{DC}$ = 1 µA).
- Min Output Power $P_{DC\_min}$ = 2.4 µW = −26 dBm.

Regarding the operating frequency, the system operates at 868 MHz in the ISM band, which offers a good balance between long-distance WPT and a small antenna form factor. This frequency band allows for the maximum permitted RF power of 27 dBm, resulting in high RF power transmission. Although lower frequencies like 433 MHz have lower free space path loss, the lower power allowed in transmission offsets this advantage, and larger antennas may be required. Higher frequencies like 2.4 GHz offer higher antenna efficiency and better form factor, but the maximum allowed transmitted power is limited to 20 dBm. Choosing the best frequency for WPT involves a trade-off between factors like allowed transmitted power, Free Space Path Loss (FSPL), and WSN size.

## 3. Results

### 3.1. MPPT Technique with Reconfigurable RF-to-DC Converter

The performance achieved by the RF-to-DC converter discussed in this work is in good agreement with targeted specifications in terms of sensitivity power level (−22 dBm) and PCE at the sensitivity power level (40%). Indeed, the RF-to-DC converter can still provide a DC output power higher than −26 dBm and, the system meets the required specification of $d_{min}$ > 7 m.

Figure 2 illustrates the schematic of a conventional three-stage CMOS Dickson rectifier, which comprises a series of cascaded voltage doublers.

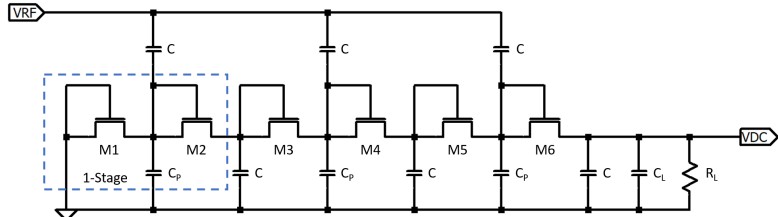

**Figure 2.** Conventional CMOS Dickson three-stage rectifier.

The circuit architecture cascades multiple rectifier stages to meet the required power sensitivity and achieve a DC output voltage across a specified load. However, this RF-to-DC converter circuitry achieves maximum PCE performance at only one input power level $P_{RF}$, which is typically very similar, if not the same, as the sensitivity power level $P_{RF\_min}$. In a typical WSN that is self-powered by an RF-to-DC converter, the SoC actively regulates the output DC voltage through the ultra-low power management to maintain a constant value of $V_{ope}$. This value is usually the maximum voltage that the rectifier devices can handle. As shown in Figure 3, maintaining a constant output voltage $V_{DC}$ (i.e., $V_{DC} = V_{ope}$) and the same number of rectifier stages $N_{oS}$ while increasing the input power $P_{RF}$ prevents the circuit from operating at its optimal condition for power conversion efficiency.

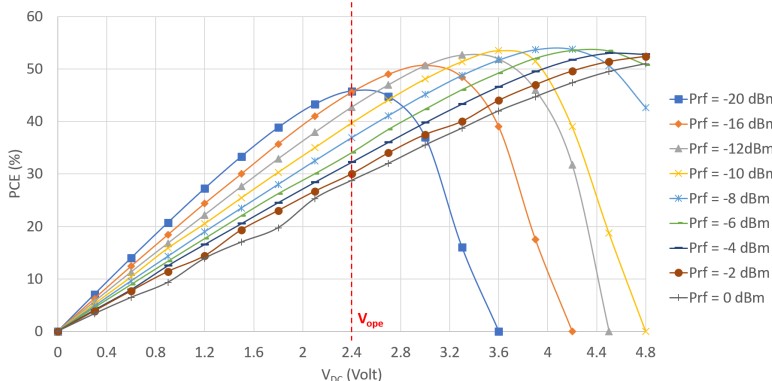

**Figure 3.** Measurements of PCE vs. DC output voltage $V_{DC}$ on a conventional six-stage CMOS Dickson rectifier at 868 MHz. $R_{load}$ = 2.2 MΩ.

The PCE performance of the conventional RF-to-DC converter is experimentally characterized in Figure 4 by varying the input power $P_{RF}$. The measurement results demonstrate that the PCE performance decreases as the input power $P_{RF}$ increases. The maximum PCE of 45% is achieved by the RF-to-DC converter at an input RF power of −20 dBm. Both Figures 3 and 4 refer to the experimental characterization of a six-stage CMOS Dickson rectifier that provides a DC output voltage $V_{DC}$ of 2.4 V when loaded with a 2.2 MΩ load resistor $R_{load}$, which is the load required to achieve the sensitivity power level. The circuit operates at a frequency of 868 MHz and, during the measurement sessions, it was directly powered through a RF power source with nine different RF power levels ranging from the sensitivity power level $P_{RF\_min}$ of −22 dBm to the maximum specified power $P_{RF\_max}$ of 0 dBm. Figure 5 illustrates that to achieve maximum sensitivity performance while optimizing the PCE, it is necessary to vary dynamically the number of converter stages $N_{oS}$ starting from the measurement of the input power $P_{in}$. This suggests the implementation of a reconfigurable RF-to-DC converter. Simulation results, later confirmed by experimental measurements, show that the best PCE performance is achieved by choosing $N_{oS}$ = 6, $N_{oS}$ = 4, and $N_{oS}$ = 3. As a result, the figure shows an RF-to-DC converter with three distinct configurations that differ in the number of rectifier stages: $N_{oS}$ = 6, $N_{oS}$ = 4, and $N_{oS}$ = 3. At the highest number of rectifier stages ($N_{oS}$ = 6), the PCE is maximum at the minimum input power ($P_{in}$ = −22 dBm). However, as the input power increases to $P_{in}$ = −14 dBm, the maximum PCE is achieved by decreasing the number of rectifier stages to $N_{oS}$ = 4. Further increasing the input power to $P_{in}$ = −10 dBm requires decreasing the number of rectifier stages to $N_{oS}$ = 3 to re-establish the highest PCE.

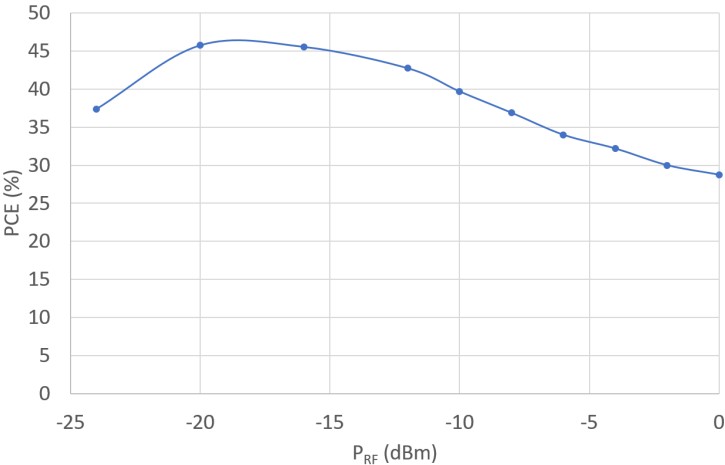

**Figure 4.** Measurements of PCE vs. $P_{RF}$ on a six-stage conventional CMOS Dickson rectifier PCE performance at 868 MHz. PCE in % vs. $P_{RF}$ in dBm. $R_{load}$ = 2.2 MΩ.

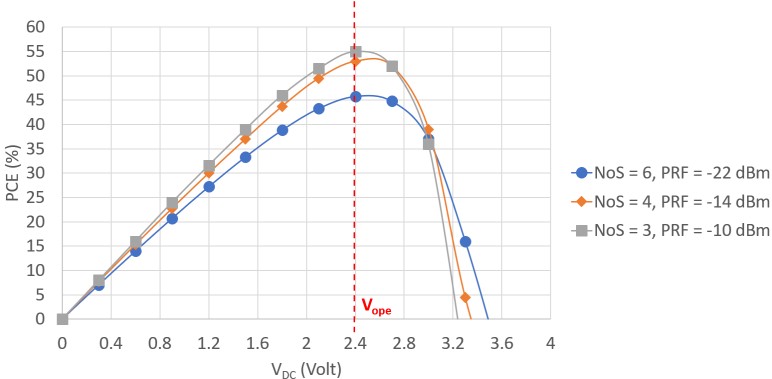

**Figure 5.** Experimental measurements of PCE vs. DC output voltage $V_{DC}$ by varying the number of stages of the RF-to-DC converter at 868 MHz. $R_{load}$ = 2.2 M$\Omega$.

The graph depicted in Figure 6 illustrates how the PCE and power sensitivity of the RF-to-DC converter vary with changes in input power, for three different configurations of the number of rectifier stages $N_{oS}$: 6, 4, and 3. The graph illustrates how the minimum RF power required to operate the RF-to-DC converter decreases with an increase in the number of rectifier stages $N_{oS}$. For instance, when $N_{oS}$ is 6, the converter produces a DC output power of 2.4 $\mu$W with a RF input power of $-22$ dBm, while $N_{oS}$ values of 4 and 3 require minimum RF input powers of $-20$ dBm and greater than $-16$ dBm, respectively, to achieve the same DC output power. The highest power sensitivity performance is achieved with the maximum number of rectifier stages, but as RF power increases, the PCE tends to decrease, making it beneficial to reduce $N_{oS}$ to 4 when $P_{RF}$ exceeds $-18$ dBm. However, when $P_{RF}$ reaches $-12$ dBm, it is advantageous to apply the technique again and reduce the number of rectifier stages to 3, resulting in a maximum PCE improvement of up to 10% compared to the six-rectifier-stage RF-to-DC converter without MPPT. Figure 7 compares the PCE performance of the RF-to-DC converter with and without the MPPT feature. The experimental characterization involved directly supplying the circuit with RF power ($P_{RF}$) that varied from $-22$ dBm to 0 dBm. The graph shows that the system with MPPT achieves $\approx 10\%$ higher PCE at peak performance while maintaining the same power sensitivity performance.

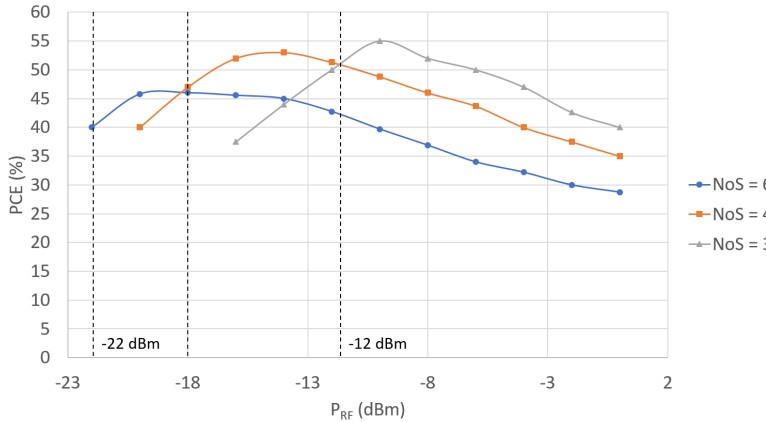

**Figure 6.** Experimental measurements of a PCE vs. $P_{RF}$ at the three different $N_{OS}$, 6, 4 and 3 at 868 MHz. $R_{load}$ = 2.2 M$\Omega$.

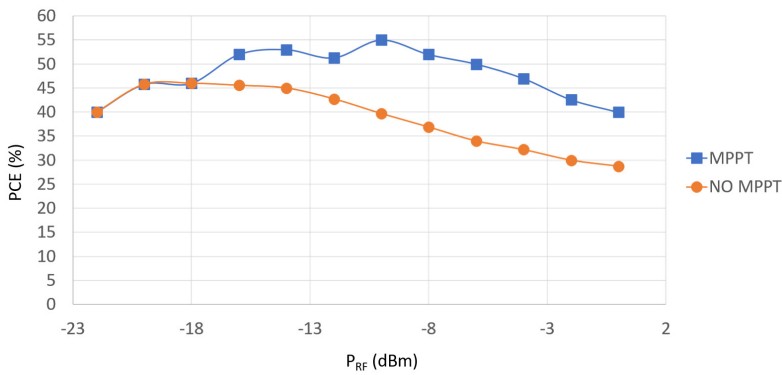

**Figure 7.** Experimental measurements of PCE vs. $P_{RF}$ on the RF−to−DC converter with and without MPPT at 868 MHz. $R_{load}$ = 2.2 MΩ.

### 3.2. MPPT System Architecture: System Design, Experimental Measurements and Validation

To perform MPPT by optimally configuring the number of rectifier stages $N_{oS}$ of the RF-to-DC converter, it is essential to continuously measure the input-received power $P_{RF}$. However, measuring $P_{RF}$ is challenging due to the need for an ultra-low-power architecture and circuitry. This work proposes an indirect and innovative method to measure $P_{RF}$ by monitoring the open-circuit voltage $V_{oc}$ at the output of the RF-to-DC converter. The relationship between the open-circuit voltage of a two rectifier stage converter and the input-received power $P_{RF}$ is illustrated in Figure 8.

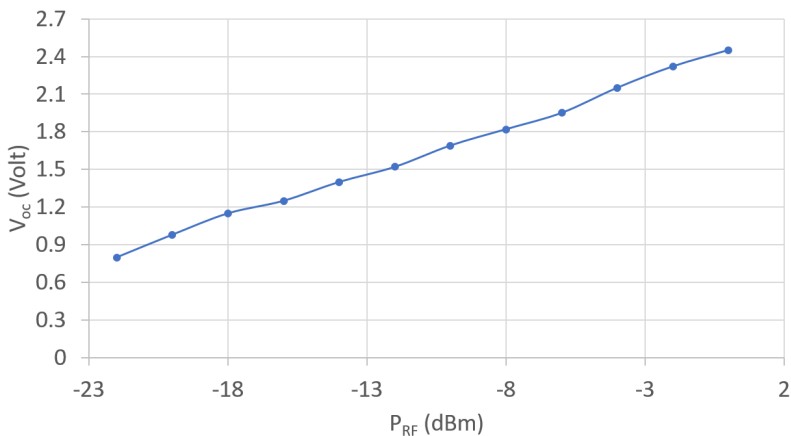

**Figure 8.** Experimental measurements of $V_{oc}$ vs. $P_{RF}$ at 868 MHz. $R_{load\_oc}$ > 10 MΩ.

As shown in the conceptual block diagram in Figure 9, the RF-to-DC converter architecture comprises two separate rectifiers connected to the same antenna to prevent energy losses that may occur when disconnecting the converter from the antenna. These rectifiers perform different functions, with one rectifier responsible for power conversion, while the other monitors and measures the RF input power through the indirect measurement of the open-circuit voltage $V_{oc}$. The RF-to-DC converter used for power monitoring is loaded with a very high impedance $R_{load\_oc}$ higher than 10 MΩ, i.e., the input gate of an NMOS transistor, it has a lower number of rectifier stages compared to the power converter stage (M < N). Using an RF-to-DC converter with a minimal number of rectifier stages has the advantages of reduced energy consumption and operation at a lower DC voltage level. The DC output of the RF-to-DC converter used for monitoring the input power is connected to a high impedance, which is the typical input stage of a CMOS comparator. Therefore, the power consumption of the power monitoring converter is significantly lower than that of the RF-to-DC power converter, and it has a negligible effect on the system PCE performance. The RF-to-DC power converter is designed with a dynamically reconfigurable number of rectifier stages $N_{oS}$ that varies as a function of the received power $P_{RF}$. It is worth high-

lighting that, with a careful layout, the RF-to-DC converter used for power monitoring and the RF-to-DC converter used for power conversion share the same PVT conditions. Indeed, across different PVT conditions, the system can track the Maximum Power Point. This aspect is an advantage of the system, as it ensures that the power monitor can accurately track the power conversion efficiency of the system, even under PVT variations.

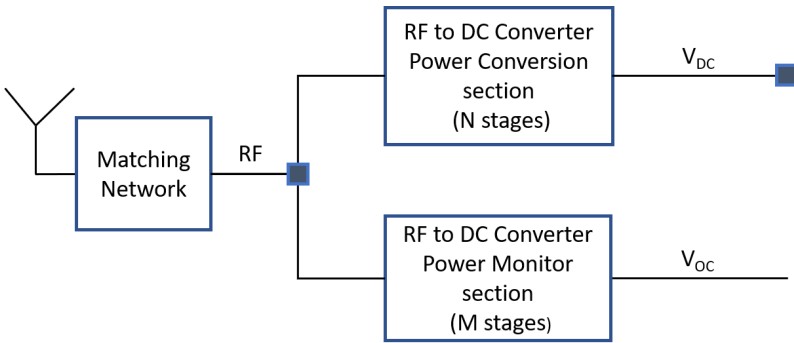

**Figure 9.** Dual RF-to-DC converter for power monitoring and power conversion.

The data presented in Figure 10 suggest that decreasing the number of rectifier stages can improve the PCE performance as the RF input power increases. Specifically, when the RF input power exceeds $-18$ dBm, the number of rectifier stages should be reduced from 6 to 4, and when it exceeds $-12$ dBm, the stages should be reduced from 4 to 3. Therefore, to determine the optimal number of rectifier stages, it is necessary to monitor the open-circuit voltage $V_{oc}$. As shown in Figure 10, to achieve an improvement in PCE performance, the number of rectifier stages must be reduced from 6 to 4 as soon as $V_{oc}$ exceeds 1.2 V ($P_{RF} > -18$ dBm), and from 4 to 3 as soon as $V_{oc}$ exceeds 1.5 V ($P_{RF} > -12$ dBm).

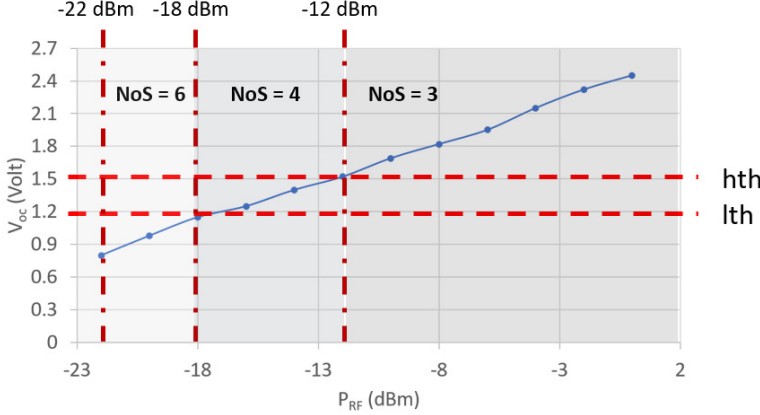

**Figure 10.** Experimental results of $V_{oc}$ and $N_{os}$ vs $P_{RF}$ at 868 MHz. $R_{load\_oc} > 10$ M$\Omega$.

As illustrated in Figure 11, an ultra-low-power voltage comparator is used to monitor the voltage $V_{oc}$. The optimum number of rectifier stages for the RF-to-DC power conversion is determined by an asynchronous Finite State Machine (FSM) through the digital signal $N_{oS}$. The FSM circuitry also manages the signals hth and lth, which define the voltage threshold provided as a voltage reference for the voltage comparator. To prevent any quiescent current due to an oscillator and affect the system performance in terms of PCE, the FSM is asynchronous and has four different states. The Power-On Reset (POR) circuitry resets the FSM when the voltage $V_{DC}$ is below 1.4 V.

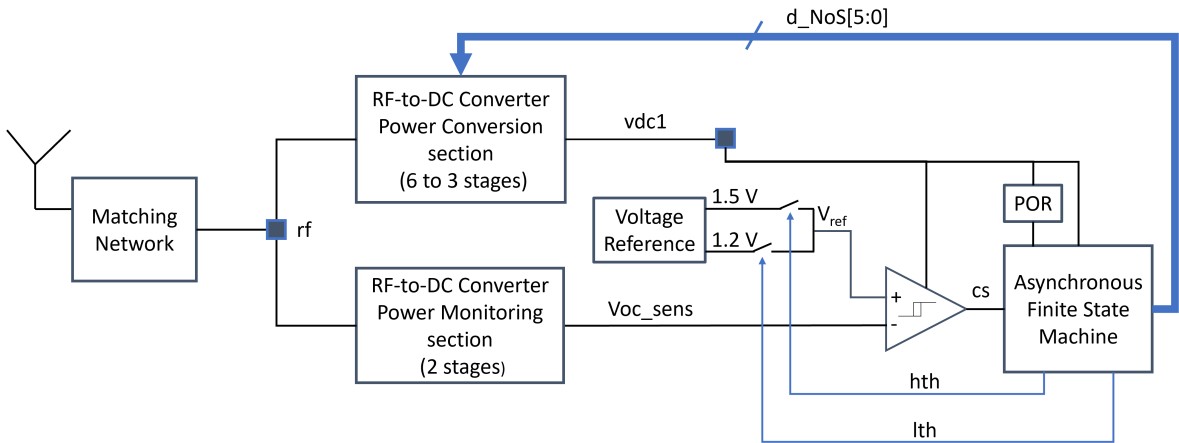

**Figure 11.** MPPT system architecture block diagram.

### 3.3. Dual RF-to-DC Converter for Power Monitoring and Power Conversion

Figure 12 shows the schematic of the dual RF-to-DC converter. The schematic shows the power monitor section implemented with a fixed 1-stage rectifier. The power conversion section is reconfigurable through the 6-bit digital bus d_NoS[5:0] provided by the FSM. The digital bus d_NoS[5:0] selects the number of stages as described in Table 1. It is worth highlighting that the 3-stage rectifier is configured by splitting into two sections, the 6-stage rectifier (M1 to M12), and reconnecting in parallel the two 3-stage sections (M1 to M6 in parallel with M7 to M12), to contribute in optimizing the PCE.

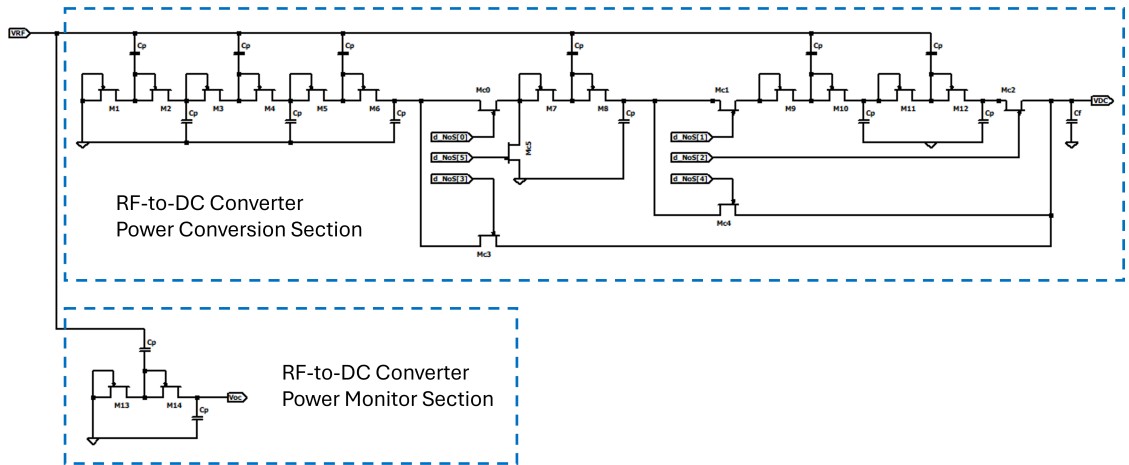

**Figure 12.** Schematic of the Dual RF-to-DC converter for power monitoring and power conversion.

**Table 1.** Digital configuration of the reconfigurable rectifier.

| d_NoS[5:0] | $N_{oS}$ | Topology |
|:---:|:---:|:---:|
| 000000 | 6 | 6-stage rectifier |
| 011010 | 4 | 4-stage rectifier |
| 101101 | 3 | 3-stage rectifier |

To ensure that the power monitor section does not affect the PCE by more than 1%, the maximum power consumption of the additional circuitry must be limited to 60 nW at 868 MHz and at the minimum $P_{RF}$ power of $-22$ dBm. Achieving this required an accurate design of the power monitoring RF-to-DC converter and a nano-power architecture of the voltage comparator. The power monitoring RF-to-DC converter has only one rectifier stage,

optimized sizes of rectifying devices, and minimized parasitic capacitances to achieve a power consumption below 50 nW when the power $P_{RF}$ is $-22$ dBm at 868 MHz. As a result, the total power consumption added by the circuitry to perform the MPPT contributes to only 1% of the 6 μW ($-22$ dBm) of the input sensitivity power $P_{RF\_min}$. This solution provides an effective advantage, as it achieves a net increase of $\approx 10\%$ in PCE performance by investing only 1% of extra power consumption. Figure 13 displays the circuit layout of the RF-to-DC converter. The picture shows that adding the power monitoring circuitry to the conventional RF-to-DC converter results in a negligible overhead in silicon area.

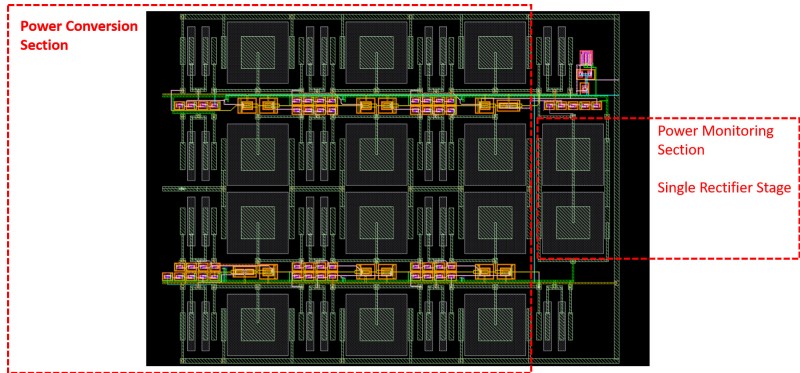

**Figure 13.** Circuit layout of the RF-to-DC evidencing the Power conversion and power monitoring sections.

### 3.4. Finite State Machine

Figure 14 depicts the FSM state diagram, which shows that the FSM enters 'State 1' after a reset. In this state, the digital bus d_NoS[5:0] is set to "000000" and configures the RF-to-DC converter to work with the maximum number of six rectifier stages. The signal $l_{th}$ is set high to establish the lower voltage threshold $V_{th\_low}$ of 1.2 V as a voltage reference to the voltage comparator, while the signal $h_{th}$ is low. The voltage $V_{oc}$ remains below $V_{th\_low}$ and the signal 'cs' is low as long as the power $P_{RF}$ remains below $-18$ dBm. When the power $P_{RF}$ exceeds $-18$ dBm, the voltage $V_{oc}$ increases beyond $_{th\_low}$, causing the signal 'cs' to go high and triggering the FSM to transition from 'State 1' to 'State 2'. In 'State 2', the digital bus d_NoS[5:0] is set to "011010" and the RF-to-DC converter operates with four rectifier stages to improve the PCE performance. The signal $l_{th}$ is low, while $h_{th}$ is high, to configure the higher voltage threshold $V_{th\_high}$ of 1.5 V as the voltage reference for the voltage comparator. When the power $P_{RF}$ exceeds $-12$ dBm, the voltage $V_{oc}$ increases beyond $V_{th\_high}$, causing the signal 'cs' to go high and triggering the FSM to transition from 'State 2' to 'State 4'. In 'State 4', the digital bus d_NoS[5:0] is set to "101101" and the RF-to-DC converter operates with three rectifier stages to improve the PCE performance. The signal $l_{th}$ is low, while $h_{th}$ is high, to configure the higher voltage threshold $V_{th\_high}$ of 1.5 V as the voltage reference for the voltage comparator. If the power $P_{RF}$ falls below $-12$ dBm while in 'State 4', the signal 'cs' goes low, and the FSM transitions from 'State 4' to 'State 3', where the RF-to-DC converter operates with four rectifier stages. The signal $l_{th}$ is high, while $h_{th}$ is low, to configure the lower voltage threshold $V_{th\_low}$ of 1.2 V as the voltage reference for the voltage comparator. If the power $P_{RF}$ falls below $-18$ dBm while in 'State 3', the signal 'cs' goes low, and the FSM transitions from 'State 3' to 'State 1'. However, if the power $P_{RF}$ exceeds $-18$ dBm, the signal 'cs' goes high, and the FSM transitions from 'State 3' to 'State 2'.

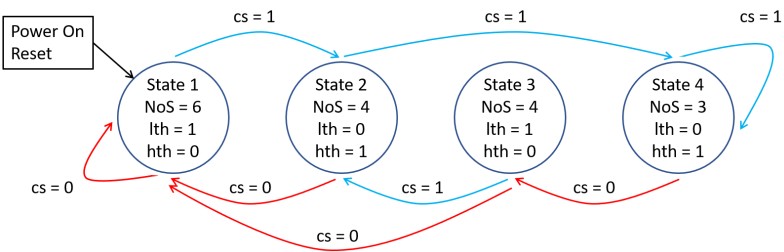

**Figure 14.** Asynchronous FSM state diagram.

### 3.5. Voltage Reference

The nano-power voltage reference is a fully CMOS voltage reference based on a self-biased topology. The circuit is resistor-less, does not require additional fabrication masks to save area and costs, and consists of only MOSFET devices operating in the subthreshold region. It is based on the well-known technique to generate two voltages with opposite temperature coefficients and add them to provide a temperature-compensated voltage with a near-zero temperature coefficient [22–24]. It implements a device in the standard 0.13 µm CMOS technology with overall current consumption of 25 nA and bias operating voltage in the range [1.1–3.3] V. Figure 15 shows the schematic of the circuit that consists of the following main sections:

- The start-up circuit consists of the transistors M18, M19, and an inverter. This circuit prevents the operation at zero current of the self-biasing circuit.
- The triode-based Widlar current reference implements an accurate temperature compensation with temperature coefficient of 100 ppm/°C. This circuit consists of the transistors M1–M6. Functionally, it is a current reference that provides a reference current to the active load.
- The active load consists of transistors M11–M17 and provides the reference voltage, $V_{ref}$ of $\approx 800$ mV.

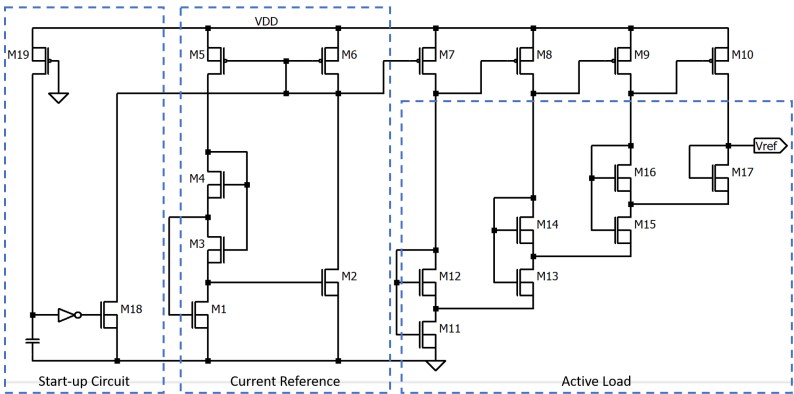

**Figure 15.** Nano-power voltage reference schematic.

The reference voltage $V_{ref}$ can be written as in (1)

$$V_{ref} = V_{th,n} + \frac{K_B \cdot T}{q} \cdot \eta \cdot \ln\left(24 \cdot \frac{K_{12} \cdot K_{14} \cdot K_{16}}{K_{11} \cdot K_{13} \cdot K_{15}} \cdot \frac{K_3 \cdot \eta^2}{\eta - 1} \cdot \ln^2\left(\frac{K_2}{K_1}\right) \cdot \left[\left(\frac{K_3 + K_4}{K_4}\right) + \sqrt{\left(\frac{K_3 + K_4}{K_4}\right)^2 - 1}\right]\right) \quad (1)$$

where

- $\eta$ is the subthreshold slope factor;
- $V_T = \frac{K_B \cdot T}{q}$ is the thermal voltage;
- $K_i = \frac{W_i}{L_i}$ is aspect ratio of transistor $M_i$;
- $W_i$ is width of transistors $M_i$;

- $L_i$ is length of transistors $M_i$;
- $V_{GSi}$ is the gate-source voltage of transistor $M_i$;
- $\mu_n$ is the carrier mobility of the n-type transistor;
- $C_{ox}$ is the gate-oxide capacitance;
- $V_{th,n}$ is threshold voltage of the n-type transistor;
- $K_B$ is the Boltzman constant;
- $T$ is the absolute temperature;
- $q$ is the elementary charge.

Equation (1) is derived as follows. The triode-based Widlar current reference implements a self-biased triode-based Widlar circuit. The transistor M3 is biased in the triode region to work like a resistor and, transistors M1 and M2 are biased in the subthreshold region. Transistors $M_5$ and $M_6$ are matched and biased in strong inversion and, in first-order approximation, their currents are equal to the reference current $I_{ref}$. The reference current is equal to the product of the drain-source voltage $V_{DS3}$ of the transistor M3 and its conductance $G_{M3}$ as given by Equation (2)

$$I_{ref} = V_{DS3} \cdot G_{M3} \tag{2}$$

$V_{DS3}$ is equal to the gate-source voltage difference of the transistors M1 and M2 as in Equation (3)

$$V_{DS3} = V_{GS1} - V_{GS2} = \eta \cdot V_T \cdot \ln\left(\frac{K_2}{K_1}\right) \tag{3}$$

$G_{M3}$ is given by Equation (4)

$$G_{M3} = \mu_n \cdot C_{ox} \cdot (V_{GS3} - V_{th,n}) \cdot K_3 \tag{4}$$

Since transistors M1 and M2 are biased in subthreshold, the drain-source current $I_{DS}$ of these transistors is as expressed in Equations (5) and (6) where, $V_{DS1}$ and $V_{DS2}$ are the respective drain-source voltages of transistors $M_1$ and $M_2$

$$I_{DS1} = \cdot\mu_n \cdot C_{ox} \cdot (\eta - 1) \cdot K_1 \cdot V_T{}^2 \cdot exp\left(\frac{V_{GS1} - V_{th,n}}{\eta \cdot V_T}\right) \cdot \left[1 - exp\left(-\frac{V_{DS1}}{V_T}\right)\right] \tag{5}$$

$$I_{DS2} = \cdot\mu_n \cdot C_{ox} \cdot (\eta - 1) \cdot K_2 \cdot V_T{}^2 \cdot exp\left(\frac{V_{GS2} - V_{th,n}}{\eta \cdot V_T}\right) \cdot \left[1 - exp\left(-\frac{V_{DS2}}{V_T}\right)\right] \tag{6}$$

Transistors $M_1$ and $M_2$ are biased with a drain-source voltage higher the $5 \cdot V_T$. This bias condition ensures that the term $\left[1 - exp\left(-\frac{V_{DS2}}{V_T}\right)\right]$ becomes negligible with a loss of accuracy in the drain-source current below 1%.

Based on this condition Equations (5) and (6) can be simplified as follows:

$$I_{DS1} \approx \mu_n \cdot C_{ox} \cdot (\eta - 1) \cdot K_1 \cdot V_T{}^2 \cdot exp\left(\frac{V_{GS1} - V_{th,n}}{\eta \cdot V_T}\right) \tag{7}$$

$$I_{DS2} \approx \mu_n \cdot C_{ox} \cdot (\eta - 1) \cdot K_2 \cdot V_T{}^2 \cdot exp\left(\frac{V_{GS2} - V_{th,n}}{\eta \cdot V_T}\right) \tag{8}$$

Since $(I_{DS1} \approx I_{DS2}) = I_{ref}$, the reference current $I_{ref}$ can be derived from Equations (2), (4), (7) and (8) as in (9)

$$I_{ref} = \underbrace{\eta \cdot V_T \cdot \ln\left(\frac{K_2}{K_1}\right)}_{V_{DS3}} \cdot \mu_n \cdot C_{ox} \cdot K_3 \cdot (V_{GS3} - V_{th,n}) \tag{9}$$

Equation (10) reports the gate-source voltage $V_{GS3}$ of transistor $M_3$

$$V_{GS3} = V_{DS3} + V_{GS4} \tag{10}$$

Transistor $M_4$ operates in strong inversion, consequently, its gate-source voltage is given by Equation (11)

$$V_{GS4} = V_{th,n} + \sqrt{\frac{2 \cdot I_{ref}}{\mu_n \cdot C_{ox} \cdot K_4}} \tag{11}$$

By combining (3), (9)–(11), the reference current can be expressed as in (12)

$$I_{ref}^2 - 2 \cdot \mu_n \cdot C_{ox} \cdot V_{DS3}^2 \cdot \frac{K_3}{K_4} \cdot (K_3 + K_4) \cdot I_{ref} + \left( \mu_n \cdot C_{ox} \cdot V_{DS3}^2 \right)^2 = 0 \tag{12}$$

By solving Equation (12), the reference current $I_{ref}$ can be expressed as in (13)

$$I_{ref} = \mu_n \cdot C_{ox} \cdot K_3 \cdot \eta^2 \cdot V_T^2 \cdot \ln^2\left(\frac{K_2}{K_1}\right) \cdot \left[ \left(\frac{K_3 + K_4}{K_4}\right) + \sqrt{\left(\frac{K_3 + K_4}{K_4}\right)^2 - 1} \right] \tag{13}$$

The reference current $I_{ref}$ is mirrored to the transistors $M_7$, $M_8$, $M_9$, and $M_{10}$ with the ratio of 1. The transistors $M_{11}$, $M_{12}$, $M_{13}$, $M_{14}$, $M_{15}$, $M_{16}$, and $M_{17}$ operate in subthreshold. The reference voltage $V_{ref}$ is given by Equation (14)

$$V_{ref} = V_{DS11} + V_{DS13} + V_{DS15} + V_{GS17} \tag{14}$$

where

$$V_{DS11} = V_{GS11} - V_{GS12} = \eta \cdot V_T \cdot \ln\left( 4 \cdot \frac{K_{12}}{K_{11}} \right) \tag{15}$$

$$V_{DS13} = V_{GS13} - V_{GS14} = \eta \cdot V_T \cdot \ln\left( 3 \cdot \frac{K_{14}}{K_{13}} \right) \tag{16}$$

$$V_{DS15} = V_{GS15} - V_{GS16} = \eta \cdot V_T \cdot \ln\left( 2 \cdot \frac{K_{16}}{K_{15}} \right) \tag{17}$$

$$V_{GS17} = V_{th,n} + \eta \cdot V_T \cdot \ln\left( \frac{I_{ref}}{\mu_n \cdot C_{ox} \cdot (\eta - 1) \cdot V_T^2 \cdot K_{17}} \right) \tag{18}$$

From equations $Vref$, $V_{DS11}$, $V_{DS13}$, $V_{DS15}$, and $V_{GS17}$, $V_{ref}$ can be expressed as in (19)

$$V_{ref} = V_{th,n} + \eta \cdot V_T \cdot \ln\left( 24 \cdot \frac{K_{12} \cdot K_{14} \cdot K_{16}}{K_{11} \cdot K_{13} \cdot K_{15}} \cdot \frac{I_{ref}}{\mu_n \cdot C_{ox} \cdot (\eta - 1) \cdot V_T^2} \right) \tag{19}$$

From (13) and (19) the reference voltage $V_{ref}$ can be written as in Equation (1).

### 3.6. Voltage Comparator

In the proposed MPPT architecture, the ultra-low-power voltage comparator operates continuously and must have a minimum quiescent power consumption. Therefore, a nano-power voltage comparator has been designed based on a common-gate architecture [25]. This design exhibits a simple circuit topology and ultra-low-power consumption, consuming below 10 nW at a bias voltage of 2.4 V. Figure 16 shows the schematic of the voltage comparator.

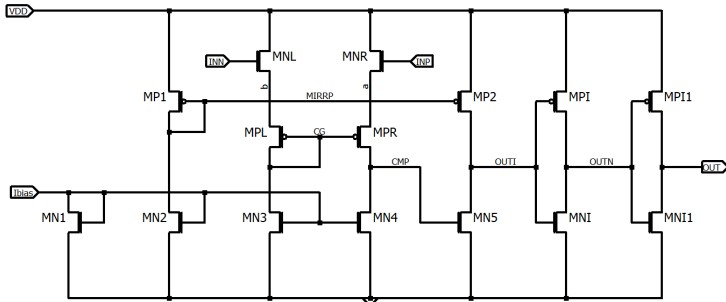

**Figure 16.** Circuit schematic of the nano-power voltage comparator.

The core of the circuit consists of the current sources, implemented with transistors MN3 and MN4, and the two pMOS transistors MPL and MPR. Transistors MN2, MN3, and MN4 have all the same aspect ratio that is double that of transistor MN1 ($K_{MN2} = K_{MN3} = K_{MN4} = 2 \cdot K_{MN1}$). The current $I_{bias}$ flowing through transistor MN1 of 0.5 nA. The two nMOS transistors MNL and MNR, are connected as source followers so that the voltages $V_a$ and $V_b$, respectively, follow the voltages $V_{INP}$ and $V_{INN}$. When the voltage $V_{INN}$ is lower than $V_{INP}$, the voltage $V_b$ is also lower than $V_a$ and the source-gate voltage of the pMOS transistor MPR is lower than that of MPL. As a consequence, the current $I_{MPR}$, which flows through the transistor MPR, is higher than the current $I_{MN4}$, which flows through the transistor MN4, so that the voltage $V_{CMP}$ goes high and, so the voltage $V_{OUT}$. When instead, $V_{INN}$ is higher than $V_{INP}$, $V_a$ is also higher than $V_b$ and the source-gate voltage of transistor MPR is higher than that of MPL. As a consequence, the current $I_{MPR}$ is lower than $I_{MN4}$ so that $V_{CMP}$ goes low and so $V_{OUT}$. Indeed, the current $I_{MPR}$ determines the voltage $V_{CMP}$ as well as the voltage $V_{OUT}$. The pMOS transistors MPL and MPR are biased in subthreshold with a source-drain voltage always higher than five times the thermal voltage $\frac{K_B \cdot T}{q}$. $K_B$ is the Boltzman constant, $T$ is the absolute temperature, and $q$ is the elementary charge. In this condition, the currents $I_{MPR}$ and $I_{MPL}$ are provided by (20) and (21), respectively. Since transistors MPR, MPL, MN3, and MN4 are biased in subthreshold with a source-drain voltage always higher than five times the thermal voltage $\frac{K_B \cdot T}{q}$, i.e., 150 mv, the minimum voltage of $V_a$ and $V_b$ cannot be lower than 300 mv so that the voltages $V_{INP}$ and $V_{INN}$ must always be higher than 700 mv.

$$I_{MPR} = \mu_p \cdot C_{ox} \cdot (\eta - 1) \cdot K_{MPR} \cdot V_T^2 \cdot exp\left(\frac{V_{SG\_MPR} - |V_{th,p}|}{\eta \cdot V_T}\right) \tag{20}$$

$$I_{MPL} = \mu_p \cdot C_{ox} \cdot (\eta - 1) \cdot K_{MPL} \cdot V_T^2 \cdot exp\left(\frac{V_{SG\_MPL} - |V_{th,p}|}{\eta \cdot V_T}\right) \tag{21}$$

From (21), the source-gate voltage $V_{SG\_MPL}$ of transistor MPL is expressed as in (22)

$$V_{SG\_MPL} = |V_{th,p}| - \eta \cdot V_T \cdot \ln\left(\frac{2 \cdot I_{bias}}{\mu_p \cdot C_{ox} \cdot (\eta - 1) \cdot K_{MPR}}\right) \tag{22}$$

The source-gate voltage of transistor MPR is expressed in (23)

$$V_{SG\_MPR} = V_b - V_{CG} = V_b - V_a + V_{SG\_MPL} \tag{23}$$

From (20) and (23), being $K_{MPR} = K_{MPL}$, the current through the transistor MPR can be expressed as in (24)

$$I_{MPR} = 2 \cdot I_{bias} \cdot exp\left(\frac{V_a - V_b}{\eta \cdot V_T}\right) \tag{24}$$

Consequently, the current out of the node CMP is given by (25)

$$I_{CMP} = I_{MPR} - 2 \cdot I_{bias} = 2 \cdot I_{bias} \cdot \left[ exp\left( \frac{V_a - V_b}{\eta \cdot V_T} \right) - 1 \right] \tag{25}$$

When $V_{INP} > V_{INN}$, which implies $V_a > V_b$, the current $I_{CMP}$ is positive and charges the gate capacitance of the transistor MN5 so that the voltage $V_{CMP}$ increases. On the contrary, when $V_{INP} < V_{INN}$, i.e., $V_a < V_b$ the current $I_{CMP}$ becomes negative and $V_{CMP}$ decreases. The voltage $V_{CMP}$ goes from its higher voltage to zero as soon as $V_a = V_b$.

### 3.7. Power Consumption Distribution

Table 2 shows the power consumption of the system sub-circuits and the relative power consumption to the specified power sensitivity $P_{RF\_min}$ and the achievable DC power consumption $P_{DC\_min}$. The table does not report the power consumption of the asynchronous FSM because it is asynchronous, so the static power consumption is zero, and only negligible dynamic power consumption should be considered during reconfiguration events.

**Table 2.** Power consumption distribution.

| | Voltage Reference | Voltage Comparator | Power Monitoring RF-to-DC Converter |
|---|---|---|---|
| Power Consumption | 100 nW | 40 nW | 50 nW |
| Relative Power Consumption to $P_{RF\_min} = 6\,\mu W\ (-22\,dBm)$ | 1.7% | 0.66% | 1% |
| Relative Power Consumption to $P_{DC\_min} = 2.4\,\mu W$ | 4.2% | 1.7% | 2% |

Table 3 provides a comparison of the system's performance with and without the MPPT implementation, as well as other state-of-the-art projects in the literature. The last two columns of the table present the PCE improvement achieved by applying the MPPT technique with the reconfigurable RF-to-DC rectifier, revealing a maximum increase of $\approx 10\%$ compared to the basic RF-to-DC rectifier without MPPT. It is relevant to note that no additional effort has been made to optimize the external matching network in this work, and a simple LC matching network tuned at 868 MHz has been designed to achieve the optimum sensitivity. Therefore, the PCE improvement obtained with this implementation serves as a reference point for a system that can be further enhanced through a reconfigurable matching network.

**Table 3.** Comparison among experimental results of the proposed RF-to-DC converter with the state-of-the-art solutions.

| Reference | [18] | [13] | [21] | [14] | This Work | This Work |
|---|---|---|---|---|---|---|
| CMOS Technology | 180 nm | 130 nm | 130 nm | 130 nm | 130 nm | 130 nm |
| Operating Frequency (MHz) | 1000 | 915 | 900 | 900 | 868 | 868 |
| PCE max (%) | 33 @ −8 dBm | 42.4 @ −16 dBm | 34.93 @ −10 dBm | 78.4 @ −16 dBm | 55 @ −12.5 dBm | 45 @ −20 dBm |
| Sensitivity (dBm) | −20.2 | −25.5 | −21.7 | −18 | −22 | −22 |
| Reconfigurable RF-to-DC Rectifier | Yes | No | Yes | No | Yes | No |
| **MPPT** | No | No | No | No | Yes | No |

### 3.8. Measurement Setup and Results

Figures 17 and 18 display the layout and microphotograph of the SoC, respectively. The SoC integrates the reconfigurable RF-to-DC converter, the MPPT system, and a RF receiver with Amplitude-Shift-Keying/Frequency-Shift-Keying (ASK/FSK) demodulation to receive data [26].

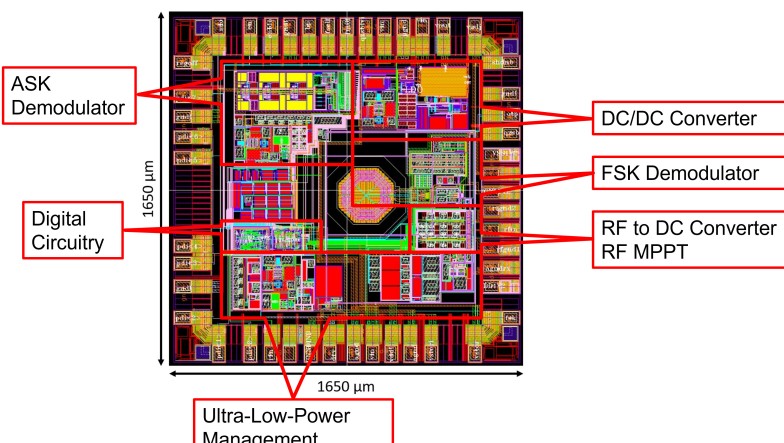

- IC area = 2.75 mm$^2$
- 0.13 μm CMOS technology (STMicroelectronics hcmos9_lprf)
- No extra Masks

**Figure 17.** SoC layout.

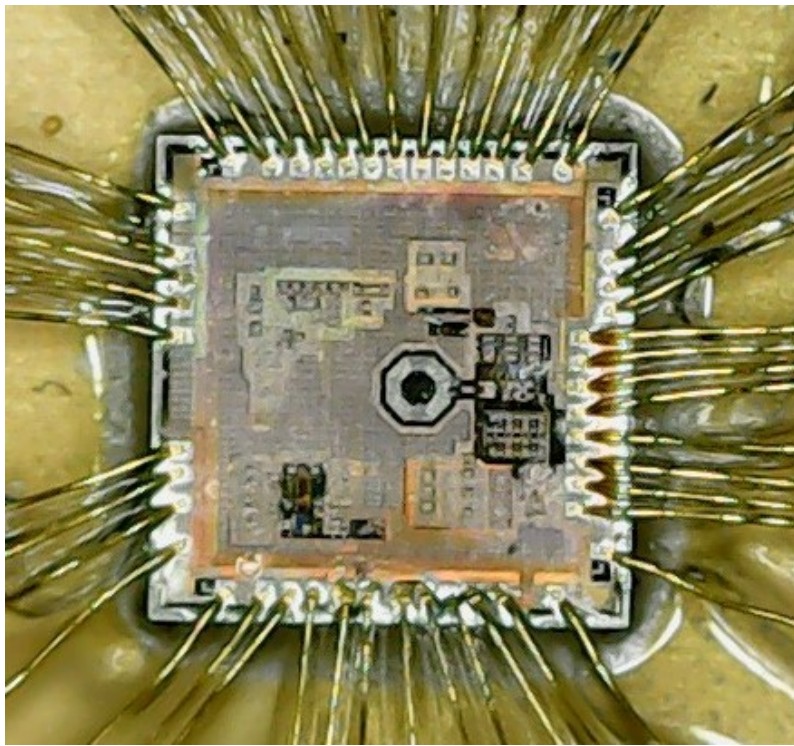

**Figure 18.** SoC microphotograph.

The system was tested in a real-world application scenario that simulates wireless power transfer in free-space between a RF reader and a power receiver that integrates the RF-to-DC converter. Figure 19 displays the two boards that were designed to implement the RF reader and the RF power receiver.

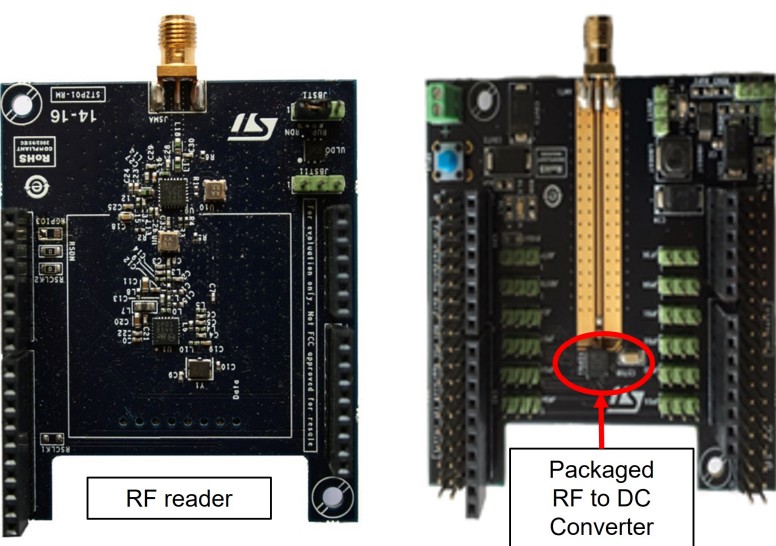

**Figure 19.** RF reader and power receiver board.

The reader utilizes STMicroelectronics' SPIRIT1, a low-power RF sub-GHz transceiver with a power amplifier that can output up to 27 dBm [27]. Both units were equipped with Laird's half-wave printed dipole antenna, specifically Revie Pro [28]. To perform wireless power transfer in free-space, the two units were placed on a measurement bench, as shown in Figure 20.

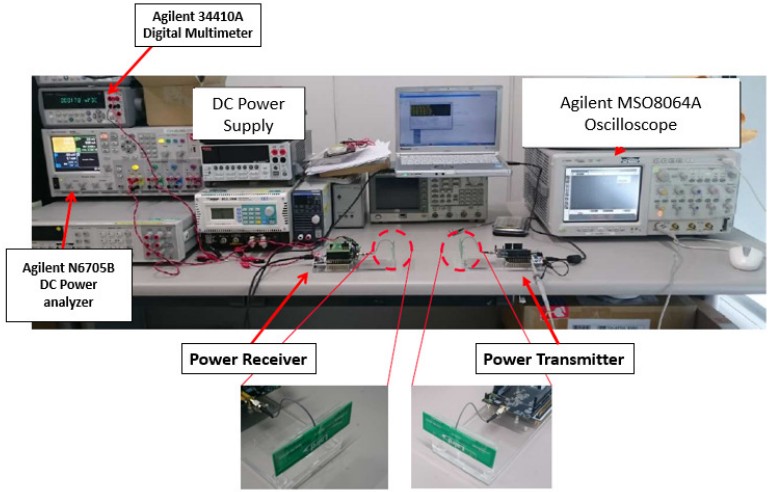

**Figure 20.** Measurement setup.

The digital multimeter 34410A from Agilent was used to measure the DC output voltage of the RF-to-DC converter, while the received power was measured using the DC power analyzer N6705B from Agilent, Santa Clara, California.

Figure 21 shows the output voltage of the RF-to-DC converter in an application where the system harvests energy into a 78 mF capacitor from a RF power source to supply a WSN. The figure shows how the RF-to-DC converter reconfigures to perform the power transfer with the highest efficiency by adjusting its stages and increasing the charging slope. At the startup, the RF-to-DC converter is configured, by default after the power on reset event, as a 6-stage converter that charges the capacitor with the minimum slope of 29 mV/s and progressively reconfigures to work as a 3-stage converter and increases the PCE, as indirectly shown by the increase of the charging slope from 29 mV/s to 42 mV/s.

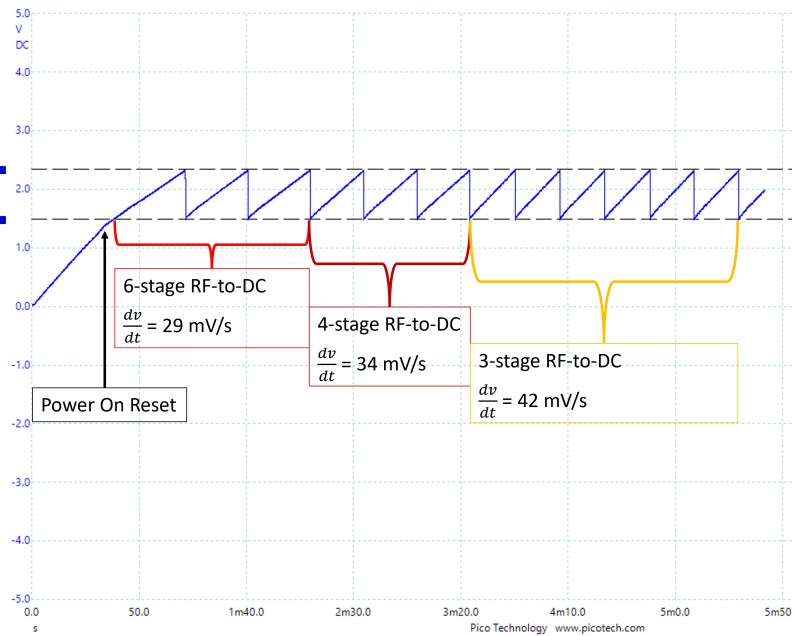

**Figure 21.** RF$-$to$-$DC power conversion at $P_{RF} = -10$ dBm.

This section may be divided by subheadings. It should provide a concise and precise description of the experimental results, their interpretation as well as the experimental conclusions that can be drawn.

## 4. Discussion

This work presented an innovative MPPT technique based on an ultra-low-power power monitoring circuitry and a reconfigurable RF-to-DC conversion circuit that showed an experimentally measured maximum PCE of 55% at 868 MHz in the input power range from $-22$ dBm to 0 dBm. Notably, this represents a 10% increase in PCE compared to a conventional RF-to-DC converter. Based on experimental results, the comparison between the PCE performance achieved using the MPPT and a conventional RF-to-DC converter was discussed. Our findings demonstrated how the new conception of the whole power conversion system achieved PCE performance improvement, regardless of the RF-to-DC converter topology, while maintaining the same performance in sensitivity. The ultra-low-power power monitoring circuitry and reconfigurable architecture of the RF-to-DC converter allowed the system to work over a wider input power range and with higher efficiency than a conventional RF-to-DC converter. Finally, this work highlighted the critical point to address in the design of ultra-low-power circuitry. It is worth noting that the same technique can be implemented in other energy sources such as photovoltaic, vibrational, and the like. In these cases, a so-conceived energy transducer would require the extra terminal for power monitoring that leads to a structure with three terminals in contrast with the state-of-the-art solution with only two terminals. This condition, at first sight, could appear as a drawback, but as revealed, this was only apparent as the induced simplicity in the WPT system architecture paid back.

**Author Contributions:** Conceptualization, R.L.R., D.D., S.C. and C.D.; methodology, R.L.R. and C.D.; SoC architecture and design, R.L.R. and C.D.; validation, R.L.R.; investigation, R.L.R., D.D., S.C. and C.D.; writing—original draft preparation, R.L.R.; writing—review and editing, R.L.R., D.D., S.C. and C.D. All authors have read and agreed to the published version of the manuscript.

**Funding:** This research received no external funding.

**Data Availability Statement:** The raw data supporting the conclusions of this article will be made available by the authors on request.

**Acknowledgments:** The authors would like to thank STMicroelectronics for providing advanced technology and expert design resources for the implementation of the application specific SoC.

**Conflicts of Interest:** Author Roberto La Rosa was employed by the company STMicroelectronics. The remaining authors declare that the research was conducted in the absence of any commercial or financial relationships that could be construed as a potential conflict of interest.

## Abbreviations

The following abbreviations are used in this manuscript:

| | |
|---|---|
| WSN | Wireless Sensor Network |
| WPT | Wireless Power Transfer |
| EH | Energy Harvesting |
| RF | Radio Frequency |
| PCE | Power Conversion Efficiency |
| MPPT | Maximum Power Point Tracking |
| MN | Matching Network |
| SoC | System on Chip |
| FSM | Finite State Machine |
| POR | Power On Reset |
| ASK | Amplitude Shift Keying |
| FSK | Frequency Shift Keying |
| FSPL | Free Space Path Loss |

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
