# Peer review of "High-Efficiency Reconfigurable CMOS RF-to-DC Converter System for Ultra-Low-Power Wireless Sensor Nodes with Efficient MPPT Circuitry"

_2674-0729, doi:10.3390/chips3010003_

Round 1

Reviewer 1 Report

Comments and Suggestions for Authors

This paper describes a Reconfigurable CMOS RF-to-DC Converter System for Ultra-Low-Power Wireless Sensor Nodes with Efficient MPPT Circuitry.

The paper needs to be revised with respect to the followings for publication.

1)     The difference from [17] should be added in more detail. [17] has two configurations whereas the authors' converter has three ones. There should be a significant difference in MPPT method.

2)     The authors need to add the load condition (Rload or Iout) for PCE – Vdc graphs of Fig. 3, 5 and the load condition (Vout, and Rload or Iout) for PCE – Prf graphs of Fig. 4, 6 – 8.

3)     If the info of Fig.7 is included in Fig. 8, Fig. 7 should be removed.

4)     Fig. 11 needs the condition of Rload.

5)     If equation (1) is disclosed in a reference, please add it. Otherwise, how to get it needs to be explained in more detail.

6)     In Fig. 17, the input voltages INN and INP need high voltages to work the comparator. An explanation about the operating condition, e.g., the minimum voltage, should be added.

Author Response

Dear Reviewer,

please find our responses in the attached file.

Kind regards,

Roberto La Rosa

Reviewer 2 Report

Comments and Suggestions for Authors

This paper presents a reconfigurable CMOS rectifier for ULP applications and MPPT circuitry. The manuscript is well-written, and presents measurements results from a fabricated chip. Some comments are:

1. Authors need to better emphasize the novelty of their work. There are many reconfigurable rectifiers in the literature, and the authors even used a very simple rectifier architecture, so the rectifier itself and the concept of reconfigurable number of stages are not new ideas. Please, clarify what is actually new.

2. What is the point of having half-page of performane metrics specifications, if in the following page it is stated that the achieved performance does not meet those specs? The reader is left with the impression that whatever follows is sub-optimal in performance.

3. The most important concern is Table 3. Most of the papers used for performance comparison are old publications (from 6 to 9 years old). Only one can be considered "recent" (4 years). Authors need to find the most-recent, state-of-the-art publications in the same topic and use it for performance comparison.

4. In Table 3, please also include the working range of input power.

Comments on the Quality of English Language

English is fine.

Author Response

Dear Reviewer,

please find our responses in the attached file.

kind regards,

Roberto La Rosa

Round 2

Reviewer 2 Report

Comments and Suggestions for Authors

The authors have adequately replied to all my comments.

No further comments are considered.